# DUDE: DEEP UNSUPERVISED DOMAIN ADAPTATION USING VARIABLE NEIGHBORS FOR PHYSIOLOGICAL TIME SERIES ANALYSIS

## ABSTRACT

Deep learning for continuous physiological time series such as electrocardiography or oximetry has achieved remarkable success in supervised learning scenarios where training and testing data are drawn from the same distribution. However, when evaluating real-world applications, models often fail to generalize due to distribution shifts between the source domain on which the model was trained and the target domain where it is deployed. A common and particularly challenging shift often encountered in reality is where the source and target domain supports do not fully overlap. In this paper, we propose a novel framework, named Deep Unsupervised Domain adaptation using variable nEighbors (DUDE), to address this challenge. We introduce a new type of contrastive loss between the source and target domains using a dynamic neighbor selection strategy, in which the number of neighbors for each sample is adaptively determined based on the density observed in the latent space. This strategy allows us to deal with difficult real-world distribution shifts where there is a lack of common support between the source and the target. We evaluated the performance of DUDE on three distinct tasks, each corresponding to a different type of continuous physiological time series. In each case, we used multiple real-world datasets as source and target domains, with target domains that included demographics, ethnicities, geographies, and/or comorbidities that were not present in the source domain. The experimental results demonstrate the superior performance of DUDE compared to the baselines and a set of four benchmark methods, highlighting its effectiveness in handling a variety of realistic domain shifts. The source code is made open-source [upon acceptance of the manuscript].

## 1 INTRODUCTION

Continuous physiological time series are important in medicine for the purpose of monitoring patient vital signs such as heart rate and blood pressure in real time. They help clinicians detect events or changes in patients condition and intervene if necessary. We defined continuous physiological time series as parameters that are recorded without interruption and with very short time intervals between samples (milliseconds or seconds). Deep learning (DL) has been successfully applied to the analysis of such time series (Perslev et al., 2019; Phan et al., 2019; Tang et al., 2022; Biton et al., 2023; Levy et al., 2023; Ribeiro et al., 2020), including the use of architectures originally developed in computer vision or natural language processing, self-supervised graph neural networks, and more. These architectures have been adapted for tasks such as sleep staging (Perslev et al., 2019; Phan et al., 2019), seizures detection (Tang et al., 2022) from the raw electroencephalogram, and the detection of arrhythmias from the electrocardiogram (ECG) (Biton et al., 2023; Ribeiro et al., 2020). DL for continuous physiological time series has been shown to equal or outperform classical machine learning models trained using handcrafted features (Levy et al., 2023; Zvuloni et al., 2023). DL has even equaled or surpassed experts in interpreting continuous physiological time series for medical tasks (Hannun et al., 2019; Ribeiro et al., 2020) or allowed them to perform tasks beyond human reach (Attia et al., 2019; Biton et al., 2021). Thus, DL has emerged as a powerful technique for achieving state-of-the-art results in various domains and applications of continuous physiological time series. However, when such models are evaluated in unseen target domains, they

tend to generalize moderately well or poorly (Alday et al., 2020; Kotzen et al., 2022; Levy et al., 2023; Ballas & Diou, 2023).

One of the key assumptions underlying traditional supervised learning approaches is that training and testing data are drawn from the same distribution. However, in real-world scenarios, this assumption may not hold. Even a slight deviation from the training domain of a network can cause it to make false predictions and significantly affect performance in the target domain where it is deployed (Tzeng et al., 2017; Long et al., 2017; Ovadia et al., 2019; Fang et al., 2020). A common and particularly challenging shift often encountered in reality is where the supports of the source and target domain do not fully overlap (Johansson et al., 2019; Sanabria et al., 2021; Tong et al., 2022).

In this work, we focus on the unsupervised domain adaptation (UDA) setting, where we assume that labeled data is available from the source domain, while only unlabeled data is available from the target domains. This setting reflects a common situation where obtaining labeled data from target domains may be costly or impractical, for example, because it requires expert clinician annotation. The underlying idea is to take advantage of the knowledge gained from the labeled data in the source domain while utilizing the unlabeled data from the target domains to align the features between the domains. A fundamental assumption we make, and which is standard in the UDA literature, is the *covariate shift* assumption. Formally, we hypothesize that the conditional distribution of the target variable given the input denoted $P_{\text{source}}(Y|X)$, remains the same in the source and target domains: $P_{\text{source}}(Y|X) = P_{\text{target}}(Y|X)$. However, the marginal distribution of the input, denoted as $P_{\text{source}}(X)$ and $P_{\text{target}}(X)$, respectively, may differ: $P_{\text{source}}(X) \neq P_{\text{target}}(X)$. This means that, while the underlying mechanisms for generating the labels remain consistent, there is a shift in the marginal distribution of the input data from the source to the target domain. In this paper, we specifically focus on deep UDA for univariate continuous physiological time series. The assumption of covariate shift is motivated by the unified medical guidelines used to generate the labels across datasets for each experiment. In this context, the medical guidelines, encapsulated by $P(Y|X)$, remain consistent across domains. However, variations in data distribution may arise due to the different populations, comorbidities, ethnicities, sensor locations, or manufacturers. Moreover, we look into real-world cases where the support of the target domain does not fully overlap the support of the source domain.

**Contributions:**

- We propose a novel framework for deep UDA, in the context of continuous physiological time series analysis. This framework is termed DUDE, an acronym for Deep Unsupervised Domain adaptation using variable nEighbors. DUDE leverages a contrastive loss between the source and target domains using a Nearest-Neighbor Contrastive Learning of Visual Representations (NNCLR) strategy (Dwibedi et al., 2021).

- We extend the original self-supervised learning (SSL) NNCLR algorithm to utilize a variable number of neighbors in the contrastive loss. The extension of NNCLR is denoted $\text{NNCLR}_\Delta$ and improves the performance of DUDE.

- We evaluated the performance of DUDE for three distinct machine learning tasks, using a total of eight datasets, and for three different types of continuous physiological time series. We demonstrate the superior performance of DUDE compared to the baselines and a set of four benchmark methods.

## 2  RELATED WORK

**Deep UDA Approaches:** Numerous deep UDA methods have been proposed in the past decade (Ganin & Lempitsky, 2015; Gulrajani & Lopez-Paz, 2021; Wilson & Cook, 2020; Shen et al., 2021; Zhou et al., 2022). These methods span various categories, such as regulation strategy, domain alignment, adversarial learning, and contrastive learning. For example, Li et al. (2022) introduces a non-Bayesian approach for modeling domain shift within a neural network. The method enhances the robustness of the model by considering uncertain feature statistics through the hypothesis of a multivariate Gaussian distribution. To address the challenge of hard sharing in representation, Amosy (2022) propose training separate models for source and target data, promoting agreement across their predictions. Dubois et al. (2021) proposed a Contrastive Adversarial Domain Bottle-

neck (CAD), with the aim of causing the encoder to produce domain-agnostic features. Supervised Contrastive Learning (SCL) (Khosla et al., 2020) improves class-wise clustering in the embedding space for the source domain.

**Contrastive learning for deep UDA:** Contrastive learning is a technique that aims to learn representations with self-supervision, ensuring that similar samples are embedded close to each other (positive pairs) while pushing dissimilar samples apart (negative pairs). Kang et al. (2019) introduced a method that minimizes the intraclass domain discrepancy and maximizes the interclass domain discrepancy. Similarly, Chen et al. (2020) proposed a framework for contrastive learning, which maximizes the agreement between the embeddings of two augmented views of the same sample and treats all other samples in the same batch as negative samples.

**Deep UDA in time series:** Ragab et al. (2022) presented a framework for UDA in the context of time series using SSL. They designed an SSL module with forecasting as an auxiliary task and proposed an autoregressive DA technique that incorporates temporal dependence of both source and target features for the purpose of domain alignment. Additionally, they trained an ensemble teacher model to align class-wise distributions in the target domain. In a different approach, Tonekaboni et al. (2021) introduced the temporal neighborhood coding algorithm, treating neighboring windows of the time series as positive pairs and using other windows to construct negative pairs, in the context of contrastive learning strategy. Contrastive learning has been used in the context of deep UDA for time series analysis by Ozyurt et al. (2023) to align the features between the source and target domains. The authors' experiments included a medical time series of measurements of vital signs collected in intensive care. However, it did not include experiments on continuous physiological time series.

**Research gap:** Deep UDA in the context of time series, particularly continuous physiological time series, remains an area of limited research. Prior to our study, Ozyurt et al. (2023) used NNCLR for the purpose of domain alignment, as one of the losses of their framework. However, to the best of our knowledge, we are the first to formally extend the NNCLR algorithm by introducing multiple neighbors into the contrastive loss and using the method in the context of continuous physiological time series. Furthermore, our work distinguishes itself by conducting a comprehensive benchmarking of our proposed DUDE framework on eight real-world datasets. These datasets encompass distribution shifts across demographics, ethnicities, geographies, and comorbidities. We evaluated DUDE on three machine learning tasks, comparing it against baselines and a set of four benchmark methods.

## 3 Proposed framework

### 3.1 Problem definition and Framework overview

We consider a supervised task (classification or regression), for which we have an univariate continuous physiological time series as input. We have several distribution shifts over different domains. The source domain, denoted $\mathcal{D}_s$, with labels i.i.d. samples given by $(x_i^s, y_i^s)_{i=1}^{N_s} \sim \mathcal{D}_s$. Furthermore, there are $\mathcal{T}$ target domains, with unlabeled samples: $\{(x_i^t)_{i=1}^{N_t} \sim \mathcal{D}_t\}_{t=1}^{\mathcal{T}}$, where $N_s$ is the number of samples in the source domain, $N_t$ is the number of samples in the $t^{th}$ target domain. Our goal is to build a model that will generalize well on the target domains $\mathcal{T}$.

The overview of the DUDE framework is shown in Figure 1. The model consists of a feature extractor responsible for processing input time series ($x^s$ or $x^t$) and generating contextual representations, denoted as $z^s$ or $z^t$. The encoder $\phi$ is task-dependent and varies according to the task. For each encoder, Domain Shift Uncertainty (DSU) layers (Li et al., 2022) are included to introduce a degree of uncertainty into the encoding process. Subsequently, the classifier $C$ leverages the encoder output to generate the desired label denoted $y$. The method is implemented in TensorFlow, version 2.6.0.

### 3.2 Domain shift uncertainty

The DSU layer was proposed by Li et al. (2022). Inside a neural network, feature statistics are treated as deterministic, i.e., a non-Bayesian approach is used: $argmax P(\theta|X) = argmax P(\theta|X)P(X)$, where $\theta$ represent the parameters of the model. The hypothesis here is that feature statistics follow a multivariate Gaussian distribution. Following this assumption, each feature vector $x$ is modeled

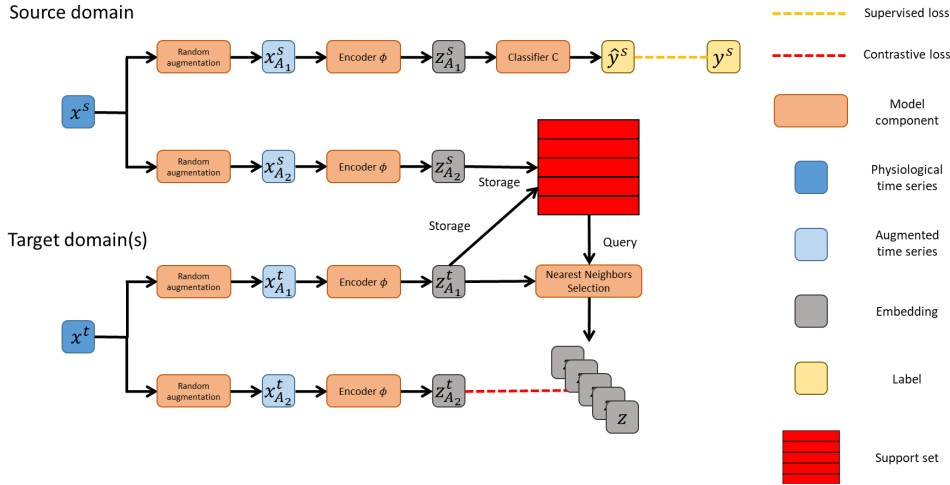

Figure 1: A high-level overview of the DUDE framework proposed. In the source domain, labeled data is used for supervised loss, which is task-dependent. The embedding generated by the encoder is stored in the support set. In the target domains, unlabeled data is used for contrastive learning. The contrastive loss with multiple nearest neighbors enables better alignment and robust DA.

as a random variable that is normally distributed, with $\mu(x)$ and $\sigma(x)$ as the mean and standard deviation of the feature statistics. Directions with larger variances can imply the potential for more valuable semantic changes. A non-parametric method is then used for uncertainty estimation: $\Sigma_\mu^2 = \frac{1}{B}\sum_{b=1}^B (\mu(x) - E_b[\mu(x)])^2$, and $\Sigma_\sigma^2 = \frac{1}{B}\sum_{b=1}^B (\sigma(x) - E_b[\sigma(x)])^2$, where $E_b$ is the empiric mean over the mini-batch $b$. Using the definition above, the DSU layer resamples the features:

$$DSU(x) = \gamma(x) \cdot \frac{x - \mu(x)}{\sigma(x)} + \beta(x) \tag{1}$$

Where $\beta(x) = \mu(x) + \epsilon_\mu \cdot \Sigma_\mu(x), \epsilon_\mu \sim \mathcal{N}(0,1)$, and $\gamma(x) = \sigma(x) + \epsilon_\sigma \cdot \Sigma_\sigma(x), \epsilon_\sigma \sim \mathcal{N}(0,1)$.

Benefiting from the proposed method, a model trained with uncertain feature statistics will gain better robustness against potential distribution shifts from the source domain and thus acquire a better generalization ability. To balance the trade-off between feature statistics enhancement and model performance, a hyperparameter denoted as $p_{DSU}$ is introduced. At each step of the training process, the re-sampling of the feature occurs with probability $p_{DSU}$. In practice, we include several DSU layers, after each principal block in the encoder.

### 3.3 NNCLR$_\Delta$ FOR DUDE

In this section, we first describe the formalism of the NNCLR algorithm, originally proposed in the context of SSL (Dwibedi et al., 2021). We then propose a method to adapt it to UDA, introducing an extension denoted NNCLR$_\Delta$, where a variable number of nearest neighbors are employed as positive instances for self-supervised contrastive learning. The nearest neighbors are selected on the basis of the cosine distance. By incorporating several neighbors, we aim to mitigate the potential noisy effects that may arise from relying solely on a single instance for positive pairing. To adapt NNCLR$_\Delta$ to the DUDE settings, we employ unlabeled data for contrastive loss calculation.

The original NNCLR algorithm (Dwibedi et al., 2021) learns self-supervised representations that go beyond single-instance positive representation, which enables the model to learn better features that are invariant to different viewpoints, deformations, and even intra-class variations. Instead of using two augmentations to form the positive pair, the nearest neighbors in the support set $Q$ are used. The support set was implemented as a queue initialized with a random matrix of size $[m, d]$, where $m$ represents the queue size and $d$ is the dimensionality of the embeddings. The support set is designed to approximate the distribution of the entire dataset in the embedding space. At the end of each training step, the support set is updated by adding the embeddings from the current batch to

the end of the queue, while discarding the oldest elements. This results in the following loss:

$$\mathcal{L}^{NNCLR} = -\log \frac{\exp\left(NN(\phi(x_i), Q)^\top \cdot \phi(x_i)^+/\tau\right)}{\sum_{k=1}^{n} \exp\left(NN(\phi(x_i), Q)^\top \cdot \phi(x_k)^+/\tau\right)} \tag{2}$$

Where $NN(\phi(x_i), Q)$ is the nearest neighbor operator, defined as $NN(\phi(x_i), Q) = \arg\min_{q \in Q} ||\phi(x_i) - q||_2$, and $\phi(x_i)^+$ means data augmentation applied on $\phi(x_i)$. In the original paper, the algorithm was proposed for image classification using augmentations such as cropping, resizing, and blurring. Instead of using a single neighbor for each sample, or even a fixed number of neighbors, NNCLR$_\Delta$ dynamically determines the number of neighbors based on the distribution of the samples in the latent space. This adaptive approach accounts for varying local densities and enhances the model's ability to handle samples from different clusters. For sample $i$, we define $NNs(x_i, Q, \Delta)$ as the set of samples in the support set $Q$ with a Euclidean distance in the latent space from $x_i$ lower than $\Delta$: $NNs(x_i, Q, \Delta) = \{x | ||\phi(x) - \phi(x_i)||_2 < \Delta, x \in Q\}$. The loss $\mathcal{L}_i$ for sample $i$ is then computed by considering all neighbors within a distance lower than $\Delta$:

$$\ell_i^u(x) = -\log \frac{\exp\left(\phi(x)^\top * \phi(x_i)^+/\tau\right)}{\sum_{j=1}^{n} \exp\left(\phi(x)^\top * \phi(x_j)^+/\tau\right)}, \tag{3}$$

$$\mathcal{L}_i^u = \frac{\sum_{a \in NNs(x_i, Q, \Delta)} \ell_i^u(a) * d(x_i, a)}{\sum_{a \in NNs(x_i, Q, \Delta)} d(x_i, a)} \tag{4}$$

where $\tau$ is a temperature parameter that controls the sharpness of the probability distribution. To account for the contributions of different neighbors, the loss is weighted based on the inverse of the Euclidean distance in the latent space between two samples, given by $d(x_1, x_2) = 1/(|\phi(x_1) - \phi(x_2)|_2)$. $\mathcal{L}_i^u$ stands for the unsupervised loss on the $i^{th}$ sample.

By dynamically selecting and weighting neighbors based on the distance threshold $\Delta$, the model adapts to the local characteristics of the data, ensuring that relevant neighbors are considered while ignoring those located far away. This adaptive mechanism improves the model's generalization capabilities in scenarios where samples are distributed across diverse clusters. Figure S2 provides a graphical intuition for $NNCLR_\Delta$.

### 3.4 TRAINING

The loss function of the DUDE framework is:

$$\mathcal{L} = \mathcal{L}^s + \lambda_u \mathcal{L}^u \tag{5}$$

Where $\mathcal{L}^s$ is the supervised loss computed on the source domain samples only. By integrating the contrastive loss with the unlabeled data and the supervised loss with the labeled data, we create a cohesive framework that enables the model to learn domain-invariant representations while preserving the discriminative information necessary for accurate classification or regression tasks. This combination ensures that the model benefits from the wealth of information present in both labeled and unlabeled data, making it well-suited for deep UDA scenarios.

## 4 EXPERIMENTS

### 4.1 EXPERIMENTS, TASKS AND DATASETS

We performed four experiments on three tasks. The experiments model clinically important tasks and use real world datasets with important distribution shifts across demographic and anthropometric (exp. 1, 4), ethnicity (exp. 2, 3), comorbidity (exp. 1), medical centers (exp. 1, 3, 4) and geography for exp. 3. In particular, there is no overlap between the source and target domains in terms of ethnicity for exp. 2, and in terms of medical centers for exp. 1, 3, 4 and geography for exp. 3. For example, the first task is the diagnosis of obstructive sleep apnea (OSA) from oxygen saturation (SpO2), which is a regression task against the apnea-hypopnea index (AHI). The second task is the detection of atrial fibrillation (AF) from the electrocardiogram (ECG). It consists of a binary

classification task. The third task is the sleep staging from photoplethysmography (PPG). It is a multiclass (four-class) classification task. For each task, we made use of a set of datasets and an original DL model previously developed (Levy et al., 2023; Ben-Moshe et al., 2022; Kotzen et al., 2022). The original source code for these three models were used. More details on the task, its clinical importance, the DL encoder used and the datasets for each experiment are provided in the supplement B.1.

**Experiment 1:** For OSA diagnosis, a total of six datasets (SHHS (Quan et al., 1997), UHV (Andrés-Blanco et al., 2017), CFS (Redline et al., 1995), Mros (Blackwell et al., 2011), and MESA (Chen et al., 2015)) were used, totaling 13,489 patients and 115,866 hours of continuous data. The first experiment considers each dataset as a domain with SHHS1 as the source domain. This experiment models important distribution shifts related to demographic and anthropometric information. The age distribution of the patients in MROS, the body mass index of CFS, are very different from the source domain (see Figure S1). The experiment also models important distribution shifts related to co-morbidity with UHV encompassing a high prevalence of patients with chronic pulmonary disease (21% in UHV versus 1% for SHHS1) (Levy et al., 2023; Quan et al., 1997) and MESA with the patients all having subclinical to clinical cardiovascular diseases (Chen et al., 2015).

**Experiment 2:** The second experiment is to consider each ethnicity as a domain. Data from all available datasets were sorted according to the ethnicity of the patient, resulting in five categories: white, Chinese American, Black and African American, Hispanic, and Asian. The white population was considered the source domain, while other ethnicities were considered as target domains. The baseline model is denoted OxiNet (Levy et al., 2023). This experiment models the effect of distribution shifts related to the ethnicity of the population sample. Historically, diagnostic algorithms were often trained on samples from the majority white population, and this experiment models this effect in the context of medical AI and evaluates the ability of deep UDA methods to tackle such effects (Norori et al., 2021). Indeed, skin pigmentation in different ethnic groups can greatly affect the oximetry signal (Visscher, 2017).

**Experiment 3:** For the diagnosis of AF, we used UVAF (Carrara et al., 2015) as the source domain and SHDB, and RBDB (Biton et al., 2023) as the target domains. The datasets total 2,047 patients and 51,386 hours of continuous ECG. The baseline model is denoted ArNet-ECG (Ben-Moshe et al., 2022). UVAF was collected in a US hospital, SHDB in a Japanese hospital, and RBDB in an Israeli hospital. This experiment models the effect of distribution shifts related to the geographical location of different hospitals. Specifically, it reflects the typical medical AI scenario where a U.S. dataset serves as the source domain (Celi et al., 2022), leading to performance drops in clinical implementation in other countries without UDA treatment.

**Experiment 4:** For sleep staging from PPG the MESA (Chen et al., 2015) was used as the source domain and CFS (Redline et al., 1995) as the test domain, totaling 2,374 patients and 23,055 hours of continuous data. The baseline model is called SleepPPG-Net (Kotzen et al., 2022). This experiment models the effect of distribution shifts related to demographics (age, see Figure S1) and of different medical centers.

## 4.2 Baseline and benchmark models

**Baselines:** In each experiment, the initial step involved training the baseline models (Levy et al., 2023; Ben-Moshe et al., 2022; Kotzen et al., 2022) through empirical risk minimization (ERM). This means training the model using ERM on the source domain data as the baseline procedure.

$$\arg\min_{\theta} \frac{1}{|\mathcal{D}|} \sum_{x,y\sim\mathcal{D}} l(f_{\theta}(x), y) \tag{6}$$

Where $\mathcal{D}$ is the source domain.

**Benchmarks:** We compared DUDE to several benchmark methods: DSU (Li et al., 2022), Coupled Training for Multi-Source Domain Adaptation (MUST)(Amosy, 2022), (CAD) (Dubois et al., 2021) and (SCL) (Khosla et al., 2020). Further details are provided in Supplements B.2.

**Data augmentation:** $NNLCR_{\triangle}$, relies on data augmentation techniques to select neighbors for contrast loss. For all tasks, Jitter augmentation is used (Saeed et al., 2019). It consists of adding white noise to the signal: $X_{new} = X + N$, where $X_{new}$ is the signal generated, $X$ is the original

signal and $N \sim \mathcal{N}(0, \sigma_{noise})$ is the noise added. For the SpO2 and PPG tasks, the Random Switch Windows ($RSW_x$) (Saeed et al., 2019) technique was used. It consists of random switching windows of $x$ seconds in the signal. For the ECG, we also used flipping and sine, as proposed in Saeed et al. (2019). Flipping consists of randomly flipping input signals with a probability of magnitude. Sine consists of adding a sine wave with random frequency and amplitude.

**Experimental settings:** For both the PPG and the ECG tasks, the recordings are divided into 30-second windows that are considered samples. For the OSA diagnosis task, a full overnight recording constitutes a sample. For each experiment, the source domain was split into train, validation, and test sets. In addition, for each experiment, a subset of each of the target domains data were used as unlabelled samples for model training. Data stratification was always performed per patient to avoid information leakage. For the OSA diagnosis and sleep staging tasks, we randomly sampled 100 patients from each target domain available to serve as the unlabeled training set. However, in the AF diagnosis task, a smaller number of patients were available. Consequently, we randomly selected only 10 patients, while the remaining 90 patients were exclusively reserved for the test set. The NNCLR$_\Delta$ algorithm also introduces one hyperparameter, $\Delta$, which defines nearest neighbors as those within the distance $\Delta$ from the query point. The training set includes labeled samples from the source domain and unlabeled samples from the target domain. The evaluation was conducted on the validation set, which consists of labeled samples exclusively from the source domain. Hyperparameter tuning was carried out using a Bayesian search with 100 iterations, meaning training the model on the train set, and validating it against the validation set. The results on the source and target domains test sets are presented for the best hyperparameter configuration found. All hyperparameters, as summarized in Table S1 were optimized.

**Performance measures:** The F1 score was used as the performance measure for exp. 1, 2, and 3 while kappa was used for exp. 4. In order to compute the confidence interval, the performance measure was repeatedly computed on randomly sampled 80% of the test set (with replacement). The procedure was repeated 1000 times and used to obtain the intervals, which are defined as follows: $C_n = \bar{x} \pm z_{0.95} * se_{boot}$, where $\bar{x}$ is the bootstrap mean, $z_{0.95}$ is the critical value found from the normal CDF distribution table, and $se_{boot}$ is the bootstrap estimate of the standard error.

## 5 RESULTS

Figure 2 presents the results of the proposed framework for all experiments. The performance of DUDE, baseline and benchmark algorithms for the OSA diagnosis task from continuous oximetry are summarized in Table 1 for the experiment per dataset (exp. 1) and in Table 2 for the experiment per ethnicity (exp. 2). For both experiments and for all datasets/ethnicities, DUDE obtained the best results. The best hyperparameter $\Delta$ found on the validation set was 0.95 for both experiments. This shows the consistency of this hyperparameter for different experimental settings. The average number of neighbors selected was 3.6. For both experiments, DUDE provided the best performance for all domains with an improvement in F1 in the range of 0.01-0.07 with respect to the baseline model for the per dataset experiment (Table 1) and 0.04-0.08 for the per ethnicity experiment (Table 2). Of note, the performance on the source domain test set was equal or superior when using DUDE (Tables 1 and 2).

The performance of DUDE, baseline, and the benchmark algorithms for the AF diagnosis task from the continuous ECG is summarized in Table 3. DUDE obtained the best results for the two target domains. The best hyperparameter $\Delta$ found on the validation set was $\Delta = 0.97$. The average number of neighbors selected was 2.8. The proposed framework outperformed the baseline for all target domains with improvement in F1 in the range 0.12-0.16 with respect to the baseline model (Table 3). Of note, the performance on the source domain test set with DUDE was equal to baseline (Table 3).

The performance of DUDE, baseline, and the benchmark algorithms for the continuous PPG sleep staging task is summarized in Table 4. For this task, DUDE outperformed the baseline as well as all benchmarks. However, the improvement over DSU, which does not utilize unlabeled data from the target domains, was incremental (0.72 for DUDE versus 0.71 for DSU). This finding suggests that, in this particular case, the assumption of covariate shift may not hold or the augmentation technique we used was not adapted. The average number of neighbors selected was 3.5. Of note, the performance on the source domain test set with DUDE was superior to the baseline (Table 4).

| | Source | | | Target | | |
|---|---|---|---|---|---|---|
| | SHHS1 | SHHS2 | UHV | CFS | MROS | MESA |
| Configuration | (562) | (505) | (257) | (486) | (3653) | (1992) |
| OxiNet (Baseline) | 0.84 | 0.83 | 0.77 | 0.78 | 0.80 | 0.75 |
| DSU (Li et al., 2022) | 0.81 | 0.81 | 0.79 | 0.79 | 0.80 | 0.77 |
| MUST (Amosy, 2022) | 0.82 | 0.81 | 0.75 | 0.77 | 0.82 | 0.78 |
| CAD (Dubois et al., 2021) | 0.80 | 0.82 | 0.78 | 0.80 | 0.80 | 0.71 |
| SCL (Khosla et al., 2020) | 0.82 | 0.83 | 0.78 | 0.75 | 0.81 | 0.76 |
| DUDE | 0.84 | 0.84 | 0.82 | 0.81 | 0.82 | 0.82 |

Table 1: Experiment 1: summary performance (F1) for the OSA diagnosis task from continuous SpO2. Per dataset experiment. The number of recordings is displayed for each domain.

| | Source | Target | | | |
|---|---|---|---|---|---|
| | white | Asian | Hispanic | BAA | Chinese |
| Configuration | (1369) | (131) | (1095) | (1487) | (245) |
| OxiNet (Baseline) | 0.79 | 0.72 | 0.71 | 0.68 | 0.66 |
| DSU (Li et al., 2022) | 0.78 | 0.75 | 0.74 | 0.70 | 0.67 |
| MUST (Amosy, 2022) | 0.79 | 0.75 | 0.70 | 0.71 | 0.69 |
| CAD (Dubois et al., 2021) | 0.74 | 0.72 | 0.73 | 0.68 | 0.67 |
| SCL (Khosla et al., 2020) | 0.76 | 0.72 | 0.71 | 0.66 | 0.71 |
| DUDE | 0.82 | 0.80 | 0.75 | 0.72 | 0.72 |

Table 2: Experiment 2: summary performance (F1) for the OSA diagnosis task from continuous SpO2. Per ethnicity experiment. The number of recordings is displayed for each domain. BAA: Black and African American

## 6 DISCUSSION

In the context of continuous physiological time series analysis, particularly in the medical domain, the presence of distribution shifts is prevalent due to various factors such as differences in data acquisition protocols, patient ethnicity, comorbidity, age, and cultural variations across healthcare institutions. These shifts can significantly impact the performance and generalization of DL models trained on a specific source domain when applied to target domains. Consequently, deep UDA becomes essential in mitigating the effects of these distribution shifts and enabling the deployment of robust and reliable models in real-world applications.

We introduced a novel framework called DUDE for deep UDA. Initially developed within the context of SSL, NNCLR was adapted for deep UDA in this work. Specifically, we incorporated a contrastive loss between samples from the source and target domains using NNCLR. Furthermore, we extended the NNCLR algorithm to accommodate a variable number of neighbors. This extension, referred to as $NNCLR_\Delta$, selects neighbors based on their proximity to the original sample in the latent space, with distances less than $\Delta$. This dynamic selection allows for varying neighbor counts per sample. The ultimate contrastive loss is computed as a weighted average across these neighbors, with weights determined by their distance from the original sample.

| | Source | Target | |
|---|---|---|---|
| | UVAF | SHDB | RBDB |
| Configuration | (258,432) | (254,592) | (205,056) |
| ArNet-ECG (Baseline) | 0.97 | 0.78 | 0.73 |
| DSU (Li et al., 2022) | 0.95 | 0.86 | 0.81 |
| MUST (Amosy, 2022) | 0.91 | 0.80 | 0.80 |
| CAD (Dubois et al., 2021) | 0.93 | 0.84 | 0.80 |
| SCL (Khosla et al., 2020) | 0.94 | 0.82 | 0.75 |
| DUDE | 0.97 | 0.90 | 0.89 |

Table 3: Experiment 3: F1 performance for the AF diagnosis task from ECG. The number of 30-second windows is displayed for each domain.

| | Source | Target |
|---|---|---|
| | MESA | CFS |
| Configuration | (2,002) | (586) |
| SleepPPG-Net (Baseline) | 0.73 | 0.70 |
| DSU (Li et al., 2022) | 0.74 | 0.71 |
| MUST (Amosy, 2022) | 0.72 | 0.71 |
| CAD (Dubois et al., 2021) | 0.72 | 0.70 |
| SCL (Khosla et al., 2020) | 0.70 | 0.66 |
| DUDE | 0.75 | 0.72 |

Table 4: Experiment 4: kappa performance for sleep staging task from PPG. The number of recordings is displayed for each domain.

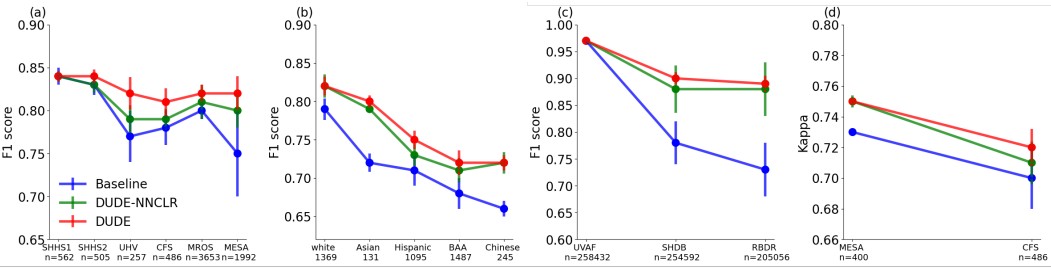

Figure 2: Results of the different experiments across the source and target domains. Error bars are produced by bootstrapping the test set. DUDE is our final framework leveraging NNCLR$_\triangle$.

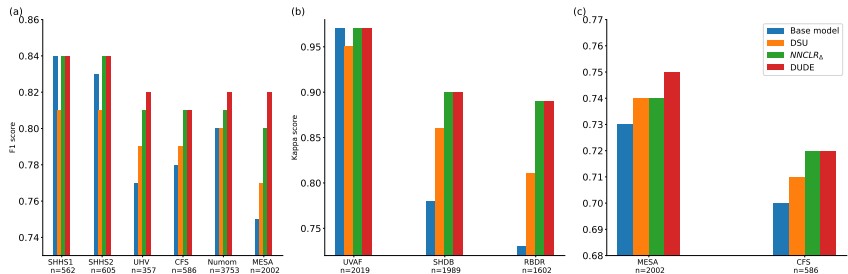

Figure 3: Ablation study on the main components of DUDE. The training of the Baseline and DSU models include the source domain training patients. The training of the NNCLR$_\triangle$ and DUDE model includes the exact same set of patients from the source domain as well as a subset of each target domain patients (as unlabeled samples). See "Experimental settings" in section 4.2.

DUDE underwent an evaluation and benchmarking on three distinct tasks: OSA diagnosis from $SpO_2$, AF diagnosis from ECG, and sleep stage detection from PPG. The DUDE framework consistently outperformed baseline algorithms across nearly all tasks and distribution shifts, as illustrated in Figure 2. We conducted a comparison of DUDE against four benchmark methods. The results consistently demonstrated DUDE's superior performance across most target domains and experiments, as indicated in Tables 1, 2, 3, and 4. Figure 2 also illustrate the improvement of DUDE using the NNCLR$_\triangle$ extension versus the original NNCLR algorithm. The ablation study (Figure 3) was performed to assess the value added by each component of DUDE. It highlights the added value of each of these components. The results show that for all experiments, the DUDE framework equals or outperforms the other benchmarks, meaning that each component has an added value. Finally, Table S2 and Table S3 show for exp.1 that the DUDE approach outperforms a strategy of randomly selecting the neighbors or when varying the number of nearest neighbors.

In the proposed NNCLR$_\triangle$ algorithm, the number of neighbors for the contrastive loss is determined on the basis of the cosine distance from the original sample. Although this approach has shown promising results, there are potential avenues for further improvement. One possible enhancement is to investigate alternative clustering techniques instead of relying solely on k-means to determine neighbors. Recent advances in clustering algorithms, such as spectral clustering (Yang et al., 2019) or hierarchical clustering (Zeng et al., 2020), offer different perspectives on identifying clusters in the latent space. Exploring these alternative clustering techniques could potentially provide more refined and accurate neighbor selection, leading to improved performance in DA scenarios. Future work should include benchmarking DUDE against additional UDA approaches such as those of Cai et al. (2021); Tonekaboni et al. (2021); Ragab et al. (2022); Eldele et al. (2023). The choice of the data augmentation techniques used in DUDE was guided by previous work (Levy et al., 2023) and by discrete experiments of common data augmentation techniques (Saeed et al., 2019) and selection with respect to their performance in the validation set. A more comprehensive and strategic approach in selecting these techniques could lead to further enhancements in DUDE's performance for a given task. Finally, although our research primarily focused on exploring the usefulness of UDA in analyzing continuous physiological time series, the DUDE framework could also be valuable for other time series UDA problems.

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
