# A SUPPLEMENTARY FIGURES AND TABLES

| Hyper-parameter | Range |
|---|---|
| $p_{DSU}$ | $[0:1]$ |
| $\Delta$ | $[0:1]$ |
| $\sigma_{noise}$ | $[1e-6:1e-4]$ |
| $RSW_x$ | task-dependant |
| $\lambda_{NNCLR}$ | $[0.1:0.5]$ |

Table S1: List of all hyper-parameters introduced by the DUDE framework.

| | Results | | | | | |
|---|---|---|---|---|---|---|
| | SHHS1 | SHHS2 | UHV | CFS | MROS | MESA |
| Configuration | (562) | (505) | (257) | (486) | (3653) | (1992) |
| DUDE | 0.84 | 0.84 | 0.82 | 0.81 | 0.82 | 0.82 |
| Random selection of neighbors | 0.74 | 0.65 | 0.61 | 0.59 | 0.63 | 0.55 |
| No supervised loss | 0.50 | 0.48 | 0.46 | 0.37 | 0.42 | 0.42 |

Table S2: Effect of randomly selecting the neighbors and inputting the supervised loss. For the Random selection of neighbors, four neighbors are used, which is close to the average number of neighbors selected with DUDE which was 3.6.

| | Results | | | | | |
|---|---|---|---|---|---|---|
| | SHHS1 | SHHS2 | UHV | CFS | MROS | MESA |
| Number of neighbors selected | (562) | (505) | (257) | (486) | (3653) | (1992) |
| 1 | 0.84 | 0.82 | 0.78 | 0.78 | 0.80 | 0.76 |
| 2 | 0.84 | 0.83 | 0.79 | 0.80 | 0.82 | 0.76 |
| 5 | 0.84 | 0.84 | 0.80 | 0.81 | 0.82 | 0.80 |
| 10 | 0.80 | 0.78 | 0.74 | 0.75 | 0.73 | 0.70 |

Table S3: Effect of number of nearest neighbors selected for DUDE-NNCLR. The results show that performance tends to increase while increasing the number from 1 to 5. Then it decreases for 10 neighbors. This observation is coherent with the intuition that leveraging the strength of incorporating a few "close" neighbors can enhance the learning process while avoiding the potential drawback of introducing too many neighbors, which could lead to increased distance from the original example.

# B SUPPLEMENTARY NOTES

## B.1 EXPERIMENTS BACKGROUND

### B.1.1 DIAGNOSIS OF OBSTRUCTIVE SLEEP APNEA FROM OXIMETRY TIME SERIES

OSA is a prevalent condition that affects up to 23.4% of women and 49.7% of men (Heinzer et al., 2015). Of particular interest for OSA diagnosis is the oximetry time series. We evaluated the ability of $NNCLR_\Delta$ to improve the generalization performance of a previously developed model, OxiNet (Levy et al., 2023).

We included five independent databases, namely SHHS (Quan et al., 1997) , UHV (Andrés-Blanco et al., 2017), CFS (Redline et al., 1995), MROS (Blackwell et al., 2011), and MESA (Chen et al., 2015). The reference AHI was defined as the average number of all apneas and hypopneas (with oxygen desaturation $> 3\%$ or arousal) per hour of sleep following the American Academy of Sleep Medicine (AASM) 2012 rules (Thornton et al., 2012), and ICSD-3 guidelines (Sateia, 2014). Signals with $\widehat{TST} < 4$ (less than 4 hours of sleep) were excluded. All remaining signals were padded to seven hours. This enables us to use signals of different lengths, from four to seven hours. After applying the exclusion criteria, all databases together include a total of 13,489 patients and 115,866 hours of continuous data. After converting the AHI into the four levels of severity i.e., non-OSA,

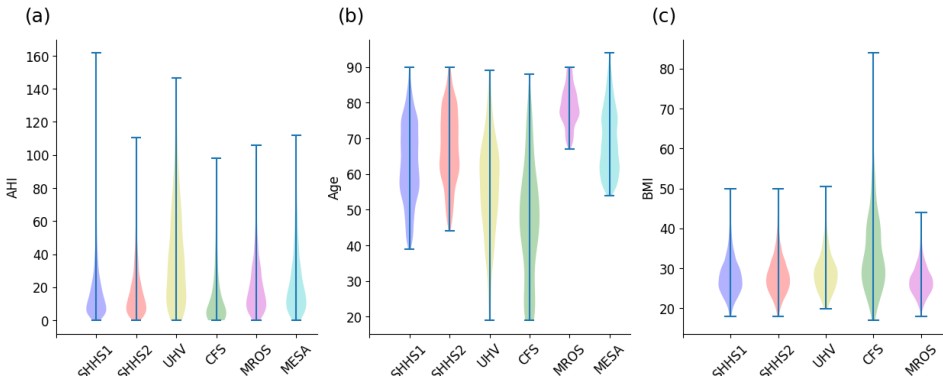

Figure S1: Violin plot for (a), Apnea Hypopnea Index (AHI), (b) Age, and (c) BMI for the study databases. BMI was not available for the MESA database. The figure is reproduced from (Levy et al., 2023) and under a Creative Commons Attribution 4.0 International License (http://creativecommons.org/).

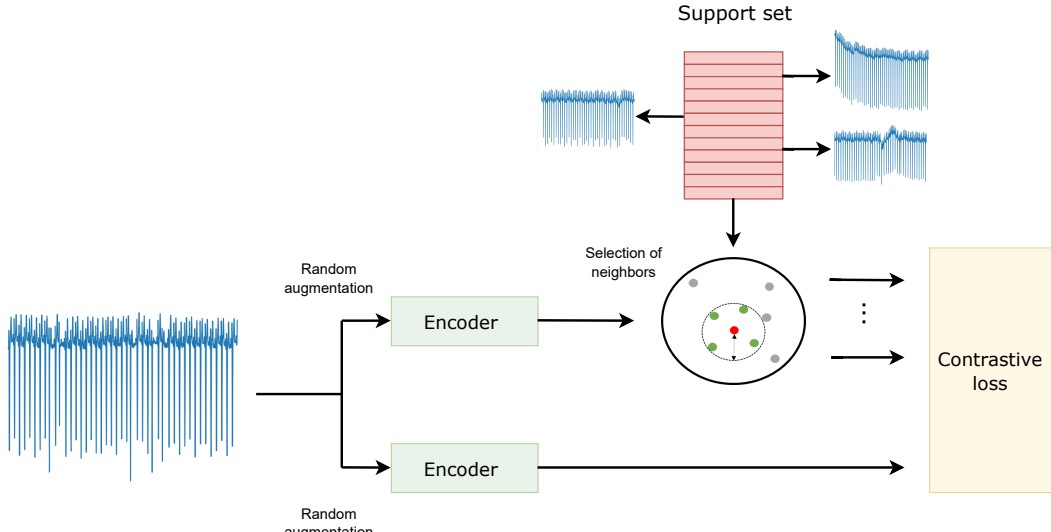

Figure S2: A high-level overview of the contrastive loss proposed in NNCLR$_\Delta$. The input signal undergoes two rounds of random augmentation, followed by embedding creation through the encoder. Neighbors are selected based on their proximity in the latent space to the first embedding, while the second one is utilized for contrastive learning.

mild, moderate, and severe OSA, the macro averaged F1 score was reported as the measure of diagnostic accuracy.

We experimented with two categories of distribution shifts. First, the hospital in which a given dataset was recorded. This actually factorizes many distribution shifts, related to the original cohort definition, the culture at a certain medical practice, and the medical equipment used for performing the sleep recordings. Indeed, (Levy et al., 2023) have shown that performance measures can drop when testing the model on a new hospital practice. To that end, the data was split according to the original datasets, resulting in six domains: SHHS1, SHHS2, UHV, CFS, MROS, and MESA. SHHS1 is considered the source domain and all other datasets are considered the target domains. Second, the ethnicity. This is particularly motivated by the fact that skin color has an influence on the measurement of the oximetry signal (Ochoa-Gutierrez et al., 2022). To that end, the data was split according to ethnicity, resulting in five categories: white, Chinese American, Black and

African American, Hispanic, and Asian. The white population was considered the source domain, while other ethnicities were considered as target domains.

Briefly, OxiNet is the base model used for this experiment. It takes as input a whole recording of oximetry, over the night. The signal is independently processed by two branches as inspired by the architecture proposed by (Interdonato et al., 2019). The first branch is based on convolutions, extracting useful patterns in the time series, called Convolutional Neural Network (CNN). The second branch, called Convolutional Recurrent Neural Network (CRNN), exploits the long-range temporal correlation present in the time series. As the AHI is the average number of desaturations per hour, RSW does not affect the label of the signal. The supervised loss used in this task is the mean squared error loss.

### B.1.2 Detection of Atrial Fibrillation from the Raw Electrocardiogram

Atrial fibrillation is the most prevalent heart arrhythmia (Björck et al., 2013). It is associated with a fivefold increase in stroke incidence and a 3.5-fold increase in mortality risk (Wolf et al., 1991). This second task focuses on evaluating the generalization performance of a previously developed model, ArNet-ECG (Ben-Moshe et al., 2022).

For this task, we have worked on three independent databases, namely UVAF (Carrara et al., 2015), SHDB, and RBDB. UVAF consists of a dataset of 2147 patients, totaling 51,386 hours of continuous ECG. UVAF consists of three-lead Holter recordings for which no lead information is available. The SHDB and RBDB datasets include 100 patients each. RBDB consists of three-lead Holter recordings (leads CM5, CC5, and CM5R), except when battery life had to be saved to handle recordings exceeding 24 h, then instead resulting in two-lead recordings (CM5 and CC5). SHDB consists of two-lead Holter ECG recordings (leads NASA and CC5). In this research the first channel of each recording was used, meaning that lead CM5 was used for RBDB and lead NASA was used for SHDB. The databases are further described in the study of (Biton et al., 2023).

The distribution shift observed in this experiment is associated with the geographic location, the type of Holter recording device used, and the placement of electrodes. The use of different recording devices and electrode placement by various hospitals can alter the morphology of the ECG signal, resulting in a shift in distribution. To that end, UVAF is considered the source domain, while SHDB and RBDB are the target domains.

The base model used for the DA experiments is named ArNet-ECG (Ben-Moshe et al., 2022). Briefly, ArNet-ECG consists of a Residual Network (ResNet) architecture (He et al., 2016) taking as input the raw ECG signal (Ben-Moshe et al., 2022). The model takes a window of 30 seconds of raw ECG as input and outputs a binary value (AF or non-AF). This base model consists of the baseline for the AF task. The supervised loss used in this task is the cross entropy loss.

### B.1.3 Sleep staging detection from continuous photoplethysmography

The final experiment conducted in this study focuses on the four-class classification of sleep staging from PPG. The four-class classification of sleep staging using PPG data holds significant motivation and potential for advancing sleep analysis and related healthcare applications. For this experiment, two databases, namely MESA (Chen et al., 2015) and CFS (Redline et al., 1995) were used. The simulated distribution shift in this experiment stems from differences in the datasets themselves, including variations in populations, recording devices, and medical practices. Since the MESA database contains a larger number of patients compared to CFS (2002 patients versus 486), it was utilized as the source domain, while CFS was designated as the target domain.

The base model employed in this experiment is SleepPPG-Net (Kotzen et al., 2022). Briefly, SleepPPG-Net is a deep learning model that takes continuous PPG time series as input. It consists of 8 stacked ResConvs, each comprising three 1D-convolutions followed by max pooling and residual addition. A windowing layer is applied after the feature extraction step to reintroduce temporal windows. These temporal windows are further compressed using a time-distributed DNN. To incorporate long-range temporal information, the model employs the Fusion Stage (FS) module, which consists of two stacked Temporal Convolutional Network (TCN) blocks. Each TCN block comprises five dilated 1D-convolutions followed by residual addition and dropout. Finally, the Fully Connected (FC) layer utilizes a 1D convolution to generate predictions. The Leaky ReLU activation

function is employed in all layers except for the output layer, which uses the Softmax activation function. The supervised loss used in this task is the categorical cross-entropy loss.

## B.2 FORMALISM OF BENCHMARKED ALGORITHMS

### B.2.1 COUPLED TRAINING FOR MULTI-SOURCE DOMAIN ADAPTATION (MUST)

Using hard sharing of representations could suffer from negative transfer because features that are useful for the source domain are emphasized over those useful for the target domain, and inference on the target data is harmed. (Amosy, 2022) proposed an alternative to hard sharing, by training separate models for source data and target data while encouraging agreement across their predictions rather than their representations. The key idea is to use the target data in supervised training, by generating pseudo-labels that agree with the target distribution. The target distribution, although through unlabeled data, contains valuable information that can be used for classification. To this end, they train two separate models. Let $N_s$ be the number of samples from the source domain, and $\{(x_i, y_i)\}_{i=1}^{N_s}$ be the labeled samples. Let $N_t$ be the number of samples from the target domain, and be $\{z_i\}_{i=1}^{N_t}$ the unlabeled samples. The teacher network is trained on labeled source samples:

$$\mathcal{L}_{teacher} = \frac{1}{N_s} \sum_{i=1}^{N_s} l_1(f_\theta(x_i), y_i) \tag{7}$$

Where $l_1$ is the loss used for the teacher optimization, and $f_\theta$ is the teacher classifier. Then the teacher trains the student using the following loss:

$$\mathcal{L}_{student} = \frac{1}{N_t} \sum_{i=1}^{N_t} l_2(f_\phi(x_i), f_\theta(x_i)) \tag{8}$$

Where $l_2$ is the loss used for the student optimization, and $f_\phi$ is the student classifier. The models are weakly coupled through their predictions, so must converge to solutions that fit both the source domain labels and the target distribution.

### B.2.2 CONTRASTIVE ADVERSARIAL DOMAIN (CAD)

Ruan et al. (Dubois et al., 2021) have introduced a method named Contrastive Adversarial Domain Bottleneck (CAD). They define $X$ as the mini-batch, and $X_{\neg D}$ samples from the target domain. They decompose the DL model as an encoder $\phi$ and a decoder $\Phi$, where the final output of the model is $\Phi(\phi(X))$. This introduces a loss:

$$\mathcal{L}_{supp} = -log \frac{\sum_{X' \in X_{\neg D}} e^{\phi(X')^\top Z}}{\sum_{X'' \in \mathbf{X}} e^{\phi(X'')^\top Z}} \tag{9}$$

where $Z = \phi(X)$. The intuition is that features extracted by the encoder should be domain-agnostic, and not contain information about the domain. This way, we minimize the mutual information of the encoding $\phi(X)$ and the target domain, normalized by the same mutual information over all the mini-batch. This loss is added to the task loss, during the optimization process. The loss $\mathcal{L}_{supp}$ is defined on both source and target domains, as no labels are needed for the target domain to derive it.

### B.2.3 SUPERVISED CONTRASTIVE LEARNING (SCL)

The SCL algorithm has been proposed by (Khosla et al., 2020). Clusters of samples belonging to the same class are pulled together in the embedding space, while simultaneously pushing apart clusters of samples from different classes.

The supervised contrastive loss is defined as follows:

$$\mathcal{L} = \sum_{i \in \mathcal{I}} \frac{-1}{|P(i)|} \sum_{p \in P(i)} log \frac{exp(\phi(x_i)^\top \phi(x_p)/\tau)}{\sum_{a \in A(i)} exp(\phi(x_i)^\top \phi(x_a)/\tau)} \tag{10}$$

Where $P(i) = \{p \in A(i) : y_p = y_i\}$ is the set of indices of all positives in the multiviewed batch distinct from $i$, and $|P(i)|$ is its cardinality. The SCL loss is added to the regression loss. As it is supervised, the loss is defined only for the source domain.

## B.3 SPACE COLLAPSE IN NNCLR$_\Delta$

To address the potential issue of latent space collapse in the NNCLR$_\Delta$ algorithm, where multiple positive instances are used for contrastive learning, it is crucial to incorporate a supervised loss on the source domain. Latent space collapse refers to a situation where the model learns to map different instances from the source and target domains into a single point or a small subspace in the latent space, resulting in a loss of discriminative power. By including the supervised loss on the source domain, which enforces class separation and preserves the discriminative information, we aim to prevent the collapse of the latent space. This combination of self-supervised contrastive learning with supervised loss provides a more robust and effective framework for DUDE, allowing the model to learn domain-invariant representations while maintaining the necessary discriminative properties for accurate classification/regression. We show that indeed imputing the supervised loss in Experiment 1 leads to very poor model performance in all target domains (Table S2).