# OpenReview forum: "DUDE: Deep Unsupervised Domain adaptation using variable nEighbors for physiological time series analysis"
_ICLR.cc/2024/Conference — Submitted to ICLR 2024_

### Official Review · Reviewer_qJ3Y · 2023-10-21

**Soundness:** 3 good
**Presentation:** 3 good
**Contribution:** 2 fair
**Rating:** 6
**Confidence:** 4

**Summary:**

This paper proposes an unsupervised domain adaptation framework for physiological time series data. The framework extends an existing Nearest Neighbor Contrastive Learning of Visual Representations (NNCLR) algorithm by allowing multiple nearest neighbors in the support set. Extensive experiments on 8 datasets with 3 tasks suggested that the model improved over existing methods on out-of-distribution domain.

**Strengths:**

1. The paper studies an underexplored area: unsupervised domain adaptation for physiological time series.
2. Experiments are extensive and results are promising.
3. Paper is easy to understand.

**Weaknesses:**

1. Some parts of the methods need more clarification or justification. For example, what are the encoders? Why is the Domain Shift Uncertainty (DSU) layer necessary?
2. Technical contribution is limited. The model extends existing NNCLR algorithm by incorporating more nearest neighbors in the support set.
3. Citation format is wrong, which reduces readability of the paper.

**Questions:**

1. What are the encoders for each task? Why is the Domain Shift Uncertainty (DSU) layer necessary? Please clarify in Methods.
2. It would be interesting to see how model performance varies across $\delta$ (i.e., number of neighbors), e.g., does model performance saturate or degrade with larger number of neighbors?
3. In ablation study (Figure S1), what’s the setup for DSU and $NNCLR_{\Delta}$? Please clarify.
4. Hidden in the supplement, the authors discuss that including a supervised loss prevents latent space collapse. Please provide an ablation study to support this claim.

---

> ### Author Response · Authors · 2023-11-23
> **Answers to Reviewer 4**
>
> We thank the reviewer for his/her appreciation of our work as well for the constructive feedback which has helped improve the manuscript.
>
> 1. Some parts of the methods need more clarification or justification. For example, what are the encoders? Why is the Domain Shift Uncertainty (DSU) layer necessary?
>
> Our answer: The encoders correspond to the original algorithms developed for a given task and published in [1,2,3]. Each encoder is briefly described in the supplementary section “B.1 experiments and background” because of the page limit for the core paper. The following sentence in the manuscript has been edited accordingly:
>
> “More details on the task, its clinical importance, the DL encoder used and the datasets for each experiment are provided in the supplement B1.”
>
> [1] Levy, Jeremy, et al. Nature Communications 14.1 (2023): 4881.
> [2] Kotzen, Kevin, et al. IEEE Journal of Biomedical and Health Informatics 27.2 (2022): 924-932.
> [3] Ben-Moshe, et al. 2022 Computing in Cardiology (CinC). Vol. 498. IEEE, 2022.
>
> The motivation for including a DSU layer is elaborated upon in question 4.
>
> 2. Technical contribution is limited. The model extends existing NNCLR algorithm by incorporating more nearest neighbors in the support set.
>
> Our answer: In this paper, we focus on the scarcely explored, yet critically important area of UDA applied to continuous physiological time series. These time series form the backbone of daily clinical practice, playing a pivotal role in the diagnosis and monitoring of patients. As researchers in this area we are confronted with a persistent, unresolved challenge in developing DL support tools for continuous physiological time series analysis: namely the effective deployment of our algorithms in real-world settings where the distribution of the data differs from the source domain. Our findings strongly support the importance of UDA in analyzing continuous physiological time series. Thus, our contribution not only enhances the algorithmic performance of NNCLR but also provides strong evidence on the potential of using UDA to bridge a crucial gap in medicine, potentially transforming patient care through more precise and adaptable diagnostic tools.
>
> 3. Citation format is wrong, which reduces readability of the paper.
>
> Our answer: We thank the reviewer for his/her remark and have now fixed the citations style.
>
> 4. What are the encoders for each task? Why is the Domain Shift Uncertainty (DSU) layer necessary? Please clarify in Methods.
>
> Our answer: We experimented with and without the DSU layer, and found that it was improving the performance based on the validation set. We thus decided to include it as one of the components of our framework. The ablation study presented in Figure 3 clearly shows that the DSU layer has a contribution to the overall DUDE performance. The DSU layer introduces some “uncertainty” in the encoder. The introduction of uncertainty makes the model less prone to overfitting on the source domain by reducing its excessive confidence, thereby facilitating domain adaptation.
>
> 5. It would be interesting to see how model performance varies across lambda (i.e., number of neighbors), e.g., does model performance saturate or degrade with larger number of neighbors?
>
> Our answer: We have explored this question for Experiment 1 and have added Table S3 where the results are reported. We ran simulations for 1, 2, 5 and 10 neighbors. The results indeed show that performance tends to increase while increasing the number from 1 to 5. However, it then decreases for 10 neighbors, i.e. performance degrades with too large a number of neighbors.
>
> 6. In ablation study (Figure S1), what’s the setup for DSU and NNCLRΔ? Please clarify.
>
> Our answer: The ablation study is now reported in the core manuscript as Figure 3. The experimental setup used is the exact same as for the main experiments conducted in the manuscript. To clarify that, we have updated the caption of Figure 3 to:
>
> “Ablation study on the main components of DUDE. The training of the Baseline and DSU models include the source domain training patients. The training of the NNCLR_Delta and DUDE model includes the exact same set of patients from the source domain as well as a subset of each target domain patients (as unlabeled samples). See "Experimental settings" in section 4.2.”
>
> 7. Hidden in the supplement, the authors discuss that including a supervised loss prevents latent space collapse. Please provide an ablation study to support this claim.
>
> Our answer: A new simulation for Experiment 1 was added to Table S2. The lack of supervised loss led to very poor model performance on all target domains. Indeed, without supervised loss, there is the issue of space collapse due to the numerous positive instances in contrastive learning. This underscores the importance of employing a supervised loss in our analysis. We added to section B.3 the reference to Table S3:

---

### Official Review · Reviewer_Pmyo · 2023-10-27

**Soundness:** 2 fair
**Presentation:** 3 good
**Contribution:** 2 fair
**Rating:** 6
**Confidence:** 4

**Summary:**

The paper proposes an unsupervised domain adaptation approach for physiological time series based on a contrastive loss that leverages nearest neighbor samples from the source domain to the target domain.

**Strengths:**

The method is simple, which is nice, and it can be used with a variety of contrastive losses, possibly with minimal changes. The evaluation is thorough across datasets, tasks, and baselines.

**Weaknesses:**

The main concern is how effective the nearest neighbor strategy is. It is the main contribution, as all the other components already exist, and unsupervised domain adaptation with self-supervised losses has been extensively studied in the vision domain.

**Questions:**

What if one selects source samples randomly instead of nearest neighbors? How many samples are really needed from the source domain to be effective for adaptation?

During adaptation, how important it is to also use a supervised loss? Ablation is very important, but it is missing.

Are the baseline results (in Table 1-4) are from corresponding papers, or are the methods reimplemented and ran by the authors?

How does the performance vary based on the changing number of nearest neighbors?

References for the augmentations used are missing. Random Switch Windows, Jitter, and Flipping were proposed in earlier work [1] on self-supervised learning for sensory data.
[1] Saeed, Aaqib, Tanir Ozcelebi, and Johan Lukkien. "Multi-task self-supervised learning for human activity detection." Proceedings of the ACM on Interactive, Mobile, Wearable and Ubiquitous Technologies 3.2 (2019): 1-30.

---

> ### Author Response · Authors · 2023-11-23
> **Answers to Reviewer 3**
>
> We thank the reviewer for his/her appreciation of our work as well for the constructive feedback which has helped improve the manuscript.
>
> 1. What if one selects source samples randomly instead of nearest neighbors? How many samples are really needed from the source domain to be effective for adaptation?
>
> Our answer: We thank the reviewer for this suggestion. We have now added a simulation for experiment 1 per which source samples are selected randomly. This experimental result is reported in Table S2 under “Random selection of neighbor”. We run the simulation for 1, 2, 5 and 10 neighbors. We also recall in Table S2 the performance of DUDE for comparison. We found that DUDE's performance was significantly and non-incrementally better to the approach of randomly selecting neighbors for all target domains. This result underscores the importance of employing a nearest neighbors approach in our analysis.
>
> Regarding the number of samples that are necessary from the source domain to enable an effective adaptation, we show in Table S3 how the number of nearest neighbors affects the performance for experiment#1. In addition, instead of working with a constant number of neighbors we have shown that there is value in using a variable number of nearest neighbors. Using NNCLR_Delta the nearest neighbors are selected on the basis of a cosine distance instead of selecting a constant number. We have shown that this dynamical approach outperforms working with a fixed number of nearest neighbors (Figure 2).
>
> 2. During adaptation, how important it is to also use a supervised loss? Ablation is very important, but it is missing.
>
> Our answer: We acknowledge the importance of an ablation study and agree that it should have been part of the main manuscript rather than the supplementary material. Accordingly, we have now included the ablation study in the main manuscript, as illustrated in Figure 3. This study highlights the added value of each critical component within the DUDE framework. Specifically, it shows the value of the NNCLR approach for the contrastive loss, the capability to vary the number of neighbors with NNCLR_Delta, and the incorporation of DSU layers into the architecture. The results show that for all experiments, the DUDE framework equals or outperforms the other benchmarks, meaning that each component has an added value.
>
> Accordingly the following sentences were added to the discussion section:
>
> “The ablation study (Figure 3) was performed to assess the value added by each component of DUDE. It highlights the added value of each of these components. The results show that for all experiments, the DUDE framework equals or outperforms the other benchmarks, meaning that each component has an added value.”
>
> Regarding the importance of using a supervised loss. A new simulation for exp. 1 was added to Table S2. The lack of supervised loss led to very poor model performance on all target domains. Indeed, without supervised loss, there is the issue of space collapse due to the numerous positive instances in contrastive learning. This underscores the importance of employing a supervised loss in our analysis. The following sentence was added to the supplementary section B.3 discussing space collapse:
>
> “We show that indeed imputing the supervised loss in Experiment 1 leads to very poor model performance in all target domains (Table S3).”
>
> 3. Are the baseline results (in Table 1-4) are from corresponding papers, or are the methods reimplemented and ran by the authors?
>
> Our answer: The encoder models from [1,2,3] and presented in Table 1-4 were obtained from the original authors. That is the exact same source code used in the original publications [1,2,3] were used for our experiments. We have added a sentence clarifying that:
>
> “The original source code for these three models were used.”
>
> [1] Levy, Jeremy, et al. Nature Communications 14.1 (2023): 4881.
> [2] Kotzen, Kevin, et al. IEEE Journal of Biomedical and Health Informatics 27.2 (2022): 924-932.
> [3] Ben-Moshe, et al. Computing in Cardiology (CinC). Vol. 498. IEEE, 2022.
>
> 4. How does the performance vary based on the changing number of nearest neighbors?
>
> Our answer: We thank the reviewer for this suggestion. We have explored this question for Experiment 1 and have added Table S3 where the results are reported. We experimented with 1, 2, 5 and 10 neighbors. The results show that performance tends to increase while increasing the number from 1 to 5. It then decreases for 10 neighbors. This observation is coherent with the intuition that leveraging the strength of incorporating a few "close" neighbors can enhance the learning process while avoiding the potential drawback of introducing too many neighbors, which could lead to increased distance from the original example.
>
> 5. References for the augmentations used are missing [...]
>
> Our answer: We thank the reviewer and have added the references provided. We have updated the corresponding text.

---

> > ### Comment · Reviewer_Pmyo · 2023-11-23
> >
> > Thanks for the clarification and additional experiments. I think the results of random baseline should be in the main paper in Table 1 or 2 and not in the appendix. I have read the author response and it addresses my concerns. I am increasing the score.

---

### Official Review · Reviewer_nstb · 2023-10-29

**Soundness:** 3 good
**Presentation:** 3 good
**Contribution:** 2 fair
**Rating:** 6
**Confidence:** 2

**Summary:**

The paper addresses the problem of domain shifts in the case of time series data and suggests a new structured approach ('DUDE') that uses a dynamic method for neighbor selection which faces the absence of
common support between the source and the target domain. The evaluation on real world datasets of continuous time series displays the higher f1 scores of the suggested framework versus the existing baselines.

**Strengths:**

- The paper develops a model and tests it on three dissimilar settings.
- Research on this domain has a considerable impact in real-world predictions of clinical interest and can be deployable as an indicative medical tool.
- The paper tries to face the obstacle of weak generalization of the trained models when used in an unobserved domain.
- It explores the realistic scenario of the training data distribution being  the testing data distribution.
- The supplement explains to some extent the logic of the loss function in figure S3, which is an incremental construction if NNCLR and seems to have emerged mostly from an empirical try.
- Although the code is missing, some of the implementation details are provided. Maybe some more would make the submission stronger.

**Weaknesses:**

- The improvement in the f1 scores is higher or same as in the existing baselines. However, only in two datasets (Target: MESA, Target: Asian) the difference form the baselines is more than 3%. This shows the encouraging consistent improvement of DUDE framework, but it is not making its superiority strong.

- Writing style: the citations are neither hyperlinked nor separated from the text.

**Questions:**

Please refer to the weaknesses.

---

> ### Author Response · Authors · 2023-11-23
> **Answers to Reviewer 2**
>
> We thank the reviewer for his/her appreciation of our work as well for the constructive feedback which has helped improve the manuscript.
>
> 1. The improvement in the f1 scores is higher or same as in the existing baselines. However, only in two datasets (Target: MESA, Target: Asian) the difference form the baselines is more than 3%. This shows the encouraging consistent improvement of DUDE framework, but it is not making its superiority strong.
>
> Our answer: The reviewer is correct that the performance of DUDE versus the baseline is very important while the performance of DUDE versus DUDE-NNCLR, although still significant, is relatively less important. It is however important to emphasize that in a healthcare diagnostic context, even a small improvement in a model’s performance can have significant implications. This enhancement can lead to more accurate and timely diagnosis, potentially saving lives by identifying diseases earlier or more accurately. In a field where precision is paramount, such improvements can significantly impact patient outcomes.
>
> 2. Writing style: the citations are neither hyperlinked nor separated from the text.
>
> Our answer: We thank the reviewer for this remark and have now fixed the citations style. Citations are hyperlinked and we use the command \citep when appropriate.

---

### Official Review · Reviewer_1WBk · 2023-10-30

**Soundness:** 2 fair
**Presentation:** 2 fair
**Contribution:** 2 fair
**Rating:** 5
**Confidence:** 4

**Summary:**

In this paper, the authors propose a Deep Unsupervised Domain adaptation using variable nEighbors (DUDE) for physiological time series analyses. Based on Nearest-Neighbor Contrastive Learning of Visual Representations (NNCLR), the authors propose a new strategy that can adaptively select the number of neighbors. Experiments on three machine learning tasks are done to verify the effectiveness of the proposed DUDE.

**Strengths:**

Strength:

1.	A DUDE framework is proposed in the context of continuous physiological time series analysis.

2.	Domain shift uncertainty (DSU) layers are applied in DUDE framework.

3.	An adaptive neighbor selection strategy is proposed.

4.	Experiments on 3 different machine learning tasks are done.

**Weaknesses:**

Weakness:

1.	The technical novelty of the work is quite limited. The two main parts of DUDE are either using existing techniques (DSU) or making marginal improvements on existing work NNCLR. Using threshold to adaptively select neighbors is not new, and it also brings a question on how to determine the threshold for different or new tasks. The paper claims that the best hyperparameter Δ found on the validation set was 0.95 for both experiments and thus shows the consistency for different experimental settings, which is not convincing. The authors may need to conduct a more comprehensive sensitivity analyses on this hyper-parameter or propose a valid hyper-parameter selection guideline for new tasks or unseen datasets.

2.	The paper highlights that DUDE is proposed for physiological time series analyses, but from the technical view, the framework can be used for general time series UDA problems. It is unclear why the context of physiological time series is necessary.

3.	Based on point 2, more general time series UDA baselines should be compared, e.g. [ref1] and [ref2], to name a few.

4.	For DSU layer, what’s the difference between DSU and instance normalization used in [ref3].

5.	Why different data augmentation used for different tasks? It is necessary to have an ablation study on different data augmentation techniques.

6.	The paper lacks ablation studies on the two parts (DSU and NNCLR) of DUDE.


[ref1] Contrastive domain adaptation for time-series via temporal mixup

[ref2] Time Series Domain Adaptation via Sparse Associative Structure Alignment

[ref3] Arbitrary style transfer in real-time with adaptive instance normalization

**Questions:**

Please refer to the weakness.

---

> ### Author Response · Authors · 2023-11-23
> **Answers to Reviewer 1**
>
> We thank the reviewer for his/her appreciation of our work as well for the constructive feedback which has helped improve the manuscript.
>
> 1. Our answer: In this paper, we focus on the scarcely explored, yet critically important area of UDA applied to continuous physiological time series. These time series form the backbone of daily clinical practice, playing a pivotal role in the diagnosis and monitoring of patients. As researchers in this area we are confronted with a persistent, unresolved challenge in developing translational DL support tools for continuous physiological time series analysis, namely the effective deployment of our algorithms in real-world settings where the distribution of the data differs from the source domain. Our findings strongly support the importance of UDA in analyzing continuous physiological time series. Thus, our contribution not only enhances the algorithmic performance of NNCLR but also provides strong evidence on the potential of using UDA to bridge a crucial gap in real-world applications of deep learning, potentially transforming patient care through more precise and adaptable diagnostic tools.
>
> With respect to the choice of the delta hyperparameter, we have further elaborated the methods section describing the hyper-parameter approach we used:
>
> “The evaluation was conducted on the validation set, which consists of labeled samples exclusively from the source domain. Hyperparameter tuning was carried out using a Bayesian search with 100 iterations, meaning training the model on the train set, and validating it against the validation set. The results on the source and target domains test sets are presented for the best hyperparameter configuration found.”
>
> In that respect we confirm that there is no information leakage in the training process of DUDE as all hyperparameters, including Δ, are learned through a search performed over the validation set.
>
> 2. Our answer: The reviewer is correct in stating that DUDE could be applied to a broad range of time series UDA problems. Given the clinical significance of continuous physiological time series and the existing gap in research on utilizing UDA for their analysis, we chose to concentrate our efforts in this area. However, we have now included a sentence in the discussion to acknowledge that the DUDE framework could indeed be valuable for analyzing other types of time series.
>
> 3. Our answer: The reviewer is correct in noting that a number of additional benchmarks could be included. Unlike other subfields of DL where SOTA approaches are well-defined, this is not the case in the field of UDA. Therefore, we have identified and included one important benchmark for each mainstream UDA approach. We then focused on demonstrating the added value of NNCLR_Delta versus NNCLR within the DUDE framework. However, we agree that expanding this research to include more UDA benchmarks is desirable. We have therefore included the following statement in the discussion section:
>
> “Future work should include benchmarking DUDE against additional UDA approaches such as those of Cai et al.(2021); Tonekaboni et al. (2021); Ragab et al. (2022); Eldele et al. (2023).”
>
> 4. Our answer: AdaIN (ref3) aligns the mean and variance of the feature from one domain to another, for the purpose of style transfer. In DSU, a non-Bayesian approach has been taken, and the goal is to insert uncertainty into the model.
>
> 5. Our answer: Specific data augmentation techniques are particularly effective for certain physiological time series, owing to the unique characteristics of the data, such as cyclic patterns and types of noise. The specific augmentation techniques chosen for each task were informed by previous experiments demonstrating the effectiveness of certain techniques for specific time series and tasks, such as the random switch for SpO2 (Levy et al 2022). This choice was also guided by discrete experiments with common data augmentation techniques (Saeed et al. 2019) and their performance evaluation on the validation set thus preventing any information leakage. The careful selection of these augmentations is crucial, as an even more strategic choice could potentially enhance DUDE's performance further and a comprehensive study benchmarking the effects of a subset of classic augmentation techniques would be beneficial. A paragraph has been added to the discussion accordingly.
>
> 6. Our answer: We acknowledge the importance of an ablation study and agree that it should have been part of the main manuscript rather than the supplementary material. Accordingly, we have now included the ablation study in the main manuscript - Figure 3. This study highlights the added value of each critical component within the DUDE framework. Specifically, it shows the value of the NNCLR approach for the contrastive loss, the capability to vary the number of neighbors with NNCLR_Delta, and the incorporation of DSU layers into the architecture.

---

### Meta-Review · Area_Chair_TCmV · 2023-12-06

**Metareview:**

The paper solves the problem of unsupervised domain adaptation through a contrastive learning framework (DUDE), tested on 3 tasks involving physiological time series. The reviewers expressed concerns about the novelty of the work, the tuning of the hyperparameter $\delta$, and the specialization of the method to physiological time series.
The authors have adequately responded to the question about hyperparameter tuning, and they also indicated that the work might be applicable to other time series. While the model does not seem to have leveraged anything that is specific to clinical data, I do not consider that the authors have an obligation to compare against other types of time series, as the their claims are that their framework works for physiological time series only. However, I do think more existing methods, developed for other types of datasets, should have been tested against. To compound the issue, the improvements upon the baselines that were tested is small, and the tables in the main paper do not include standard deviations, so it's difficult to tell how consistent these improvements are just from the tables. Figure3 includes error bars, which makes things clearer, however, I wonder how realistic the splitting by race is for a real dataset. It looks like the expected improvement compared to the baseline varies a lot by target dataset, and it is still not clear to me why this happens.
Ultimately, I do not judge this paper's methodological contribution to be sufficient for ICLR, neither do I find the insights generalizable beyond these specific datasets/tasks.

**Justification For Why Not Higher Score:**

The paper's contributions are limited both in terms of novelty and empirical evaluation.

**Justification For Why Not Lower Score:**

N/A

---

### Decision · Program_Chairs · 2024-01-16

Reject